# Polyester nasal swabs collected in a dry tube are a robust and inexpensive, minimal self-collection kit for SARS-CoV-2 testing

**Leah R. Padgett[1], Lauren A. Kennington[1], Charlotte L. Ahls[1], Delini K. Samarasinghe[1], Yuan-Po Tu[2], Michelle L. Wallander[3], Shawna D. Cooper[4], James S. Elliott[1], Douglas Rains[1] \***

1 Quantigen Biosciences, Fishers, Indiana, United States of America, 2 The Everett Clinic-Part of Optum, Everett, Washington, United States of America, 3 Sciest, Mooresville, North Carolina, United States of America, 4 Audere, Seattle, Washington, United States of America

\* doug.rains@quantigen.com

**Data Availability Statement:** All relevant data are within the manuscript and its Supporting information files.

## Abstract

### Background

In order to identify an inexpensive yet highly stable SARS-CoV-2 collection device as an alternative to foam swabs stored in transport media, both contrived ("surrogate") CoV-positive and patient-collected spun polyester swabs stored in dry tubes were evaluated for time- and temperature-stability using qPCR.

### Methods

Surrogate specimens were prepared by combining multiple, residual SARS-CoV-2-positive clinical specimens and diluting to near-LOD levels in either porcine or human mucus ("matrix"), inoculating foam or polyester nasal swabs, and sealing in dry tubes. Swabs were then subjected to one of three temperature excursions: (1) 4°C for up to 72 hours; (2) 40°C for 12 hours, followed by 32°C for up to 60 hours; or (3) multiple freeze-thaw cycles (-20°C). The stability of extracted SARS-CoV-2 RNA for each condition was evaluated by qPCR. Separate usability studies for the dry polyester swab-based HealthPulse@home COVID-19 Specimen Collection Kit were later conducted in both adult and pediatric populations.

### Results

Polyester swabs stored dry demonstrated equivalent performance to foam swabs for detection of low and moderate SARS-CoV-2 viral loads. Mimicking warm- and cold- climate shipment, surrogate specimens were stable following either 72 hours of a high-temperature excursion or two freeze-thaw cycles. In addition, usability studies comprised of self-collected patient specimens yielded sufficient material for molecular testing, as demonstrated by RNase P detection.

**Funding:** Funding for this project was provided primarily by the Bill and Melinda Gates Foundation, through a direct award (BMGF COVID Grant Investment ID: INV-016831) to Quantigen Biosciences (LRP, LAK, CLA, DKS, JSE, and DR) and Audere (SDC). The funder provided support in the form of salaries for authors (LRP, LAK, CLA, DKS, MLW, SDC, JSE, and DR) but did not have any additional role in the study design, data collection and analysis, decision to publish or preparation of the manuscript. The specific roles of these authors are articulated in the 'author contributions' section. Some reagents and consumables were provided free of charge to Quantigen by ThermoFisher. LRP, LAK, CLA, DKS, JSE, and DR are employed by Quantigen. YT is employed by The Everett Clinic-Part of Optum and MLW is employed by Sciest LLC.

**Competing interests:** The authors have read the journal's policy and have the following competing interest: LRP, LAK, CLA, DKS, JSE, and DR are employed by Quantigen Biosciences. YT is employed by The Everett Clinic-Part of Optum and MLW is employed by Sciest LLC. These affiliations do not alter our adherence to PLOS ONE policies of sharing data and materials. There are no patents, products in development or marketed products associated with this research to declare.

## Conclusions

Polyester nasal swabs stored in dry collection tubes offer a robust and inexpensive self-collection method for SARS-CoV-2 viral load testing, as viral RNA remains stable under conditions required for home collection and shipment to the laboratory.

## Introduction

The worldwide outbreak of severe acute respiratory syndrome coronavirus-2 (SARS-CoV-2) has caused unprecedented demand for testing to diagnose patients with coronavirus disease 2019 (COVID-19). To expand capacity and increase accessibility, nasal (anterior nares, AN) specimens were previously evaluated for SARS-CoV-2 molecular testing and were deemed acceptable by the FDA, leading to their inclusion in recommendations by both the CDC and the FDA (https://www.fda.gov/medical-devices/coronavirus-covid-19-and-medical-devices/faqs-testing-sars-cov-2#testingsupply) [1]. Recent studies also demonstrated that self-collected AN specimens are equivalent to healthcare professional-collected nasopharyngeal and oropharyngeal swabs for detection of SARS-CoV-2 by reverse transcriptase-polymerase chain reaction (RT-qPCR) [1, 2]. Nasal swab self-collection is more tolerable and reduces patient contact with medical professionals, thereby decreasing the risk of viral transmission and preserving PPE supplies.

In early March of 2020, the initial studies which established the nasal swab collection method utilized foam swabs [1, 2]. However, supplies quickly became depleted as large diagnostic manufacturers, laboratories, and government agencies secured the available supply. To address the shortage, additional swab types were evaluated including spun polyester swabs, which can be manufactured at higher capacity than foam swabs and are inexpensive to produce. The polyester spun swab demonstrated similar performance for nasal specimen collection to the foam swab, and FDA guidance was updated to include polyester swabs as an acceptable swab type for SARS-CoV-2 testing [3].

Upper respiratory swabs have historically been transferred to the laboratory in viral transport media (VTM); however, VTM shortages ensued upon the start of the pandemic. Phosphate buffered saline (PBS), saline, and other alternative solutions were deemed acceptable by the FDA for swab transport, with saline demonstrating better performance for use with nasal swabs as compared to VTM [3]. While alternate swab transport solutions eventually helped to alleviate acute supply shortages, laboratories have continued to explore additional swab transport methods for the purposes of both controlling costs and obviating future supply chain disruptions.

In this study, we evaluated the performance and stability of polyester nasal swabs stored in dry collection tubes as an alternative specimen collection option for SARS-CoV-2 molecular testing. Initial data were gathered from replicate "surrogate" (i.e., contrived) swabs prepared by diluting residual positive clinical specimens in human or porcine matrix to concentrations near the qPCR assay's reported limit of detection (LoD). The conclusions of these studies were confirmed by the assessment of fresh positive patient specimens collected with polyester nasal swabs. We first demonstrate that surrogate specimens remain stable when stored dry and subjected to temperatures that mimic warm- and cold-climate shipment. Subsequent usability studies in healthy pediatric and adult populations confirm the feasibility of at-home, nasal specimen self-collection.

## Materials and methods

### SARS-CoV-2 clinical specimens for creating surrogate specimens

Swab specimens used to prepare "surrogate" SARS-CoV-2-positive samples for this study were provided by the Everett Clinic (Everett, WA; UnitedHealth Group protocol #2020–0002, approved by UHG Institutional Review Board). These residual patient samples had known positive or negative SARS-CoV-2 status, as determined by previous qPCR testing with the Thermo Fisher EUA-approved TaqPath COVID-19 Combo Kit [3].

### Creation of surrogate specimens

At Quantigen, residual negative patient samples were pooled and diluted 1:2 in porcine mucus (the latter obtained from two independent sources: the University of Minnesota Veterinary Diagnostic Lab and the Iowa State University Veterinary Medicine Laboratory). The resulting human-porcine matrix, both somewhat viscous and highly unsterile, served as the "cold pool"–i.e., free of SARS-CoV-2 infection—for preparing CoV-negative surrogate samples. In parallel, twelve high-positive SARS-CoV-2 patient specimens (Ct values < 25 cycles for all three viral target genes–*ORF1ab*, *N*, *S*—when tested at Quantigen) were combined to generate a master positive, known colloquially as a "hot pool." Replicate qPCR wells of this starting hot pool were then tested with the Thermo assay to determine average Ct values. For pinpointing the desired concentrations, target Ct values corresponding to 2x LoD (20 genomic copy equivalents (GCE)/PCR reaction) and 10x LoD (100 GCE/PCR reaction) were mathematically determined using the formula $2^{-\Delta Ct} = F$, where $\Delta Ct$ is the difference in expected Ct between the 1x LoD (*N* gene mean Ct 29.1, according to the Thermo EUA filing) and the 2x and 10x LoD pools; F is the fold change in each case [4]. Using the empirically determined *N* gene values from several replicate qPCR wells, depending on the experiment, the master hot pool was diluted with an appropriate volume of negative human-porcine matrix or negative-human only matrix to generate the 2x LoD (low-positive) and 10x LoD (high-positive) hot pools, respectively. Once these two dilutions of the first hot pool were performed, ten replicates of the resulting 2x LoD and 10x LoD hot pools were qPCR-tested to confirm that *N* gene Ct values were within the expected range (± 20% linear fold change). SARS-CoV-2 GCE values for the hot pools were determined by *N* gene standard curve analysis and digital PCR (S1 Text).

Once concentrations were verified, the 2x and 10x LoD hot pools were separately spiked onto replicate nasal swabs. For thoroughness, several brands of swab were included: Puritan foam (Puritan Medical Products #25–1506), Fisherbrand polyester-tipped applicator (Fisher Scientific #22-363-170), Copan spun polyester (Copan Diagnostics #164KS01), SteriPack spun polyester (SteriPack #60564, also known as US Cotton spun polyester #3), SteriPack spun polyester (SteriPack #60567, also known as US Cotton spun polyester 3ARS), Copan mini FLOQ (Copan Diagnostics #503CS01), and Copan regular FLOQ (Copan Diagnostics #56380CS01). Swabs were stored dry in sealed 15 mL screw top tubes. Sterile saline (Kroger) was used for swab storage when media was utilized.

### Dry swab elution

After the appropriate temperature excursions, described in a later section, elution of material from the dry swabs was achieved by adding 1 mL of PBS (Sigma) to the 15 mL screw top tube, performing a 30 second vortex with intermittent pulsing, and incubating for 10 minutes at room temperature.

## Nucleic acid extraction and RT-qPCR

Nucleic acid extraction was performed from 400 μL PBS eluate using the MagMAX™ Viral/Pathogen Nucleic Acid Isolation Kit (Thermo Fisher). RT-qPCR was performed on the QuantStudio™ 7 Flex 96-well Fast (Thermo Fisher) using 5 μL RNA and the TaqPath COVID-19 Combo Kit according to the EUA Instructions For Use, 14March2020; Pub. No. MAN0019181 Rev. A.0 (https://www.thermofisher.com/content/dam/LifeTech/Documents/PDFs/clinical/taqpath-COVID-19-combo-kit-full-instructions-for-use.pdf). Raw cycle thresholds were obtained using the QuantStudio™ Real-Time PCR Software v1.3 and the following thresholds: 40,000 for *ORF1ab*, 25,000 for *N*, and 38,000 for *S*. Samples with indeterminate or spurious amplification signals were designated a cycle threshold (Ct) value of 40 cycles. For the experiments using human matrix, samples were also tested using the TaqMan® RNase P assay (Thermo Fisher) on the QuantStudio™ Dx Real-Time Instrument.

## Viral recovery from swabs

Viral recovery from polyester swabs (Copan) was evaluated across a range of viral loads (1x to 64x LoD) by serial dilution of the master hot pool with negative clinical matrix. The viral pools were tested directly (no swab) or spiked onto polyester swabs. To create the spiked swabs, polyester swabs were submerged into a 1.5 mL tube containing 40 μL of hot pool and abraded against the tube wall five times in each direction to mimic nasal collection or until all of the specimen material was absorbed. The inoculated swab was then transferred to a 15 mL screw cap tube. For the direct viral testing, 40 μL of hot pool was aliquoted into a 15 mL screw cap tube. Three replicates were performed for each of the hot pools and conditions (+/- swab). All tubes, with or without swabs, were incubated for 30 minutes at room temperature followed by dry swab elution with 1 mL PBS for the spiked swab or addition of 1 mL PBS directly to the tube (no swab).

For the evaluation of alternative swab elution methods, paired polyester swabs (SteriPack #60567) were collected from the AN of 10 healthy volunteers by an equal loading method with a crossover design [5]. Swabs were then spiked with 40 μL of the low-positive (2x LoD) hot pool and stored at 4˚C in a dry tube overnight. Swabs were divided into two groups, each containing five pairs, and eluted by the addition of 1 mL PBS. Group 1 compared the standard vortex method with swab swirling (swirling the swab within the tube for 10 seconds) while group 2 compared the standard vortex method to passive elution for 1 hour under refrigeration. RNA was extracted for SARS-CoV-2 and *RNase P* testing by RT-PCR.

## Evaluation of surrogate specimen stability (porcine matrix)

A two-arm study (S1 Fig) was performed to evaluate the stability of surrogate/contrived clinical SARS-CoV-2-positive specimens prepared on foam (Puritan) and polyester swabs (Fisherbrand) stored in dry tubes, as well as on polyester swabs (Fisherbrand) stored in a tube containing 1 mL of sterile saline. Swabs were stored at either 4˚C (control arm) throughout, or at 40˚C for the initial 12 hours and then held at 32˚C (experimental arm). To create the surrogate specimens, the 2x LoD and 10x LoD hot pools described earlier were added directly to the swabs through a procedure that mimicked nasal swabbing action. Each swab was placed in a 1.5 mL microcentrifuge tube containing 80 μL of the appropriate hot pool, with the bud submerged into the mixture and abraded in a circular motion against the side of the tube. After 30 seconds, each hot pool-saturated swab was dropped and sealed in a 15 mL screw cap tube containing either 1 mL saline or no media whatsoever ("dry"). After inoculation, baseline (time 0) positive and negative swabs from the control arm were processed immediately. Remaining swabs were stored at 4˚C or 40˚C for the control and experimental arms, respectively. After 12

hours at 40˚C, experimental swabs were transferred to 32˚C. Viral stability for both saline- and dry-stored swabs was assessed at three time points: 1) 24 hours (+ 8 hours), 2) 48 hours (+ 8 hours), and 3) 72 hours (+ 8 hours). Note that the additional eight hours were included as an FDA requirement for stability labeling. Swabs were retrieved from storage two hours prior to each time point to equilibrate to room temperature.

## Evaluation of surrogate–SARS-CoV-2 plus human matrix and clinical specimen stability

Dry swab specimen stability using freshly collected, human-only matrix was evaluated through 72 hours at 4˚C and through 48 hours at elevated temperatures (40˚C 12 hours, held at 32˚C). Healthy volunteers were each given one Copan polyester swab and one SteriPack #60564 polyester swab and were instructed to collect normal AN human matrix onto the swabs to generate roughly equivalent replicates as follows: 1) five revolutions of the Copan swab (left nostril), 2) five revolutions of the SteriPack #60564 swab (right nostril), 3) five revolutions of the same SteriPack #60564 swab (left nostril), and 4) five revolutions of the same Copan swab (right nostril) [5]. The swabs were collected from volunteers within an eight-hour timeframe and stored overnight in 15 mL screw cap tubes at 4˚C prior to viral inoculation. Each swab was placed in a 1.5 mL microcentrifuge tube containing 40 μL of one SARS-CoV-2 high-positive (10x LoD) patient sample, with the tip submerged into the mixture and abraded in a circular motion against the side of the tube for 30 seconds or until complete absorption. Swabs were transferred to dry 15 mL screw cap tubes. Baseline (time 0) swabs were processed and tested immediately, while all remaining swabs were stored at 4˚C or 40˚C for the control and experimental arms, respectively. After 12 hours at 40˚C, experimental swabs were transferred to 32˚C. Specimens were retrieved from storage two hours prior to each time point and were allowed to equilibrate to room temperature. Specimen stability was assessed at 24 hours (+ 8 hours) and 48 hours (+ 8 hours) in both study arms and at 72 hours (+ 8 hours) in the control arm.

For the evaluation of clinical specimens, paired SteriPack polyester #60564 swabs were collected from five SARS-CoV-2 positive and five SARS-CoV-2 negative patients by the same method described above. Paired dry swabs were stored at 4˚C or 40˚C for the control and experimental arms, respectively. After 12 hours at 40˚C, experimental swabs were transferred to 32˚C. Specimens were retrieved from storage two hours prior to each time point and were allowed to equilibrate to room temperature. Specimen stability was assessed at 72 hours (+ 8 hours).

## Evaluation of dry polyester swab freeze-thaw stability

For the evaluation of freeze thaw stability, paired SteriPack polyester #60567 swabs were collected from ten healthy volunteers by the same method described above. The swabs were collected from volunteers within an eight-hour timeframe and stored overnight in 15 mL screw cap tubes at 4˚C prior to viral inoculation. Each swab was placed in a 1.5 mL microcentrifuge tube containing 40 μL of high-positive (10x LoD) hot pool, with the bud submerged into the mixture and abraded in a circular motion against the side of the tube for 30 seconds or until complete absorption. Swabs were transferred to dry 15 mL screw cap tubes and stored at 4˚C (control) or -20˚C (experimental) for 18 hours. All experimental swabs were then incubated at room temperature for 4 hours (one freeze thaw). This process was repeated for a second freeze thaw cycle (-20˚C for 20 hours; 4 hours at room temperature). Control swabs remained at 4˚C until processing.

## Self-collection kit usability studies

Usability studies for the HealthPulse@home COVID-19 Specimen Collection Kit (Audere) were performed in adult and pediatric populations to determine if individuals (or parent/ guardians) could successfully complete a nasal swab collection procedure and ship the specimens to a lab without professional assistance. Healthy adult (ages 18–90 yr.) and pediatric (ages 3–17 yr.) participants were recruited into separate studies using targeted online advertisements (Audere protocols #Pro00044524 and #Pro00046977, approved by the Advarra Institutional Review Board). Enrollment was monitored to ensure that the distribution of demographic factors within the study sets such as gender, race/ethnicity, and education level approximated the US population. Exclusion criteria were recent respiratory infection, recent nasal surgery, prior experience with nasal swab collection, prior medical or laboratory training, pre-existing coagulopathy, and use of anticoagulant medications. Eligible participants were consented online and sent a home collection kit with instructions to not open the kit prior to their scheduled video session. During the recorded observational video session, participants opened the kit to access both paper and step-by-step digital instructions. The latter were accessible by a QR code leading to a registration website. Participants followed the directions to register their kit and perform an anterior nares swab using one polyester nasal swab (SteriPack #60567) for both nostrils, place the swab into a dry tube, and prepare the specimen kit for shipment to a lab. Shipping choices included dropping off the kit at a FedEx drop box, scheduling a pick-up via phone, or scheduling a pick-up online. After completing the specimen collection procedure, participants were surveyed to assess the "ease of use" of the collection kit. Acceptance criteria was that no more than 15% of participants should be determined by the observational researcher as having "A lot of difficulty" on any single step. Based on feedback from the adult study, kit registration and ship-back instructions were modified for the pediatric study.

Upon arrival at the lab, kits were inspected for packaging errors and specimens were deemed acceptable or unacceptable for testing. Dry nasal specimens were eluted and tested for RNase P as described above. The acceptance criteria for specimen collection within each study population was a mean *RNase P* Ct value of $\leq 32$ across all specimens (based on conversations with FDA during informal pre-submission discussions regarding emergency use authorization for the Audere home collection kit).

## Data analysis

Specimen stability at each time point was defined as $\geq 95\%$ agreement with the expected result for low-positive samples (2x LoD) and $\geq 100\%$ agreement with the expected results for high-positive (10x LoD) and negative control samples. When calculating the differences between Ct values, negative ΔCt values indicate a lower Ct and thus higher viral load, while positive ΔCt values indicate a higher Ct value and thus lower viral load. All statistical analyses (one-way ANOVA and t-test) were performed using GraphPad Prism 8 (GraphPad Software).

# Results

## Swab absorption and elution from swabs stored in dry format

Surrogate specimens were prepared by spiking polyester and foam nasal swabs with SARS-CoV-2 positive "hot" pools (Fig 1A) using a swab swirling procedure to mimic nasal swabbing. Polyester swabs were 150% more absorbent than foam swabs by the swirling method and by passive absorption (Table 1) and thus, polyester swabs may be advantageous in the clinical setting as they contain a larger sample capacity. Once prepared, surrogate specimens were stored dry in empty tubes and eluted with 1 mL PBS, followed by a 30 second vortex and 10 minute

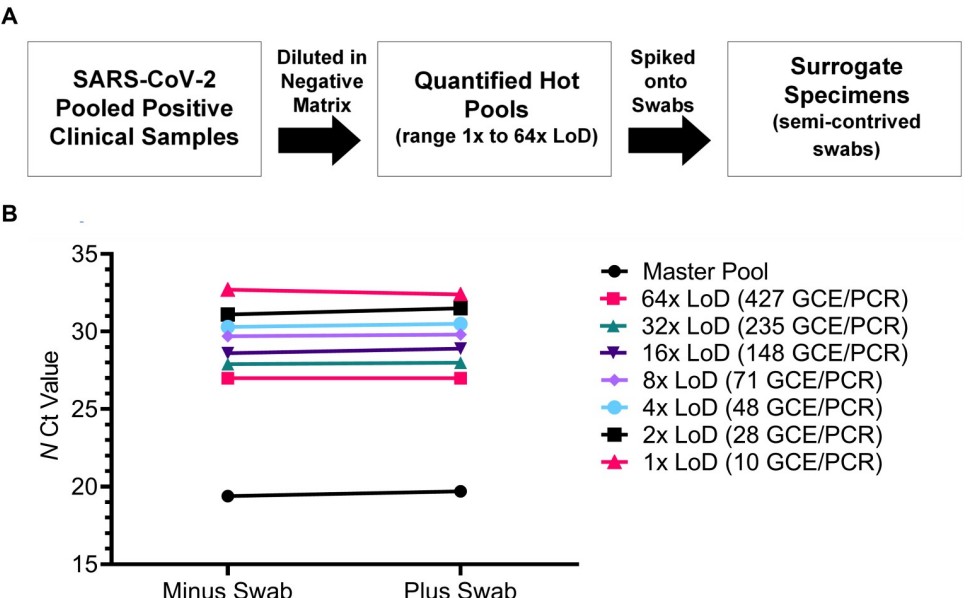

**Fig 1. Absorption and recovery of specimens from dry polyester swabs.** (A) Dry polyester SARS-CoV-2 surrogate specimens were created across a range of viral loads (1x to 64x LoD). (B) The absorption and recovery of SARS-CoV-2 from dry polyester surrogate specimens was evaluated. *N* gene mean Ct values (n = 3 replicates) from hot pool serial dilutions tested directly (minus swab) were compared to Ct values from swabs spiked (plus swab) with hot pool serial dilutions.

room temperature incubation. Specimen recovery from dry swabs was evaluated by comparing Ct values from serially diluted surrogate specimens created with polyester swabs to Ct values from serially diluted hot pools tested directly, without absorption and elution from swabs. SARS-CoV-2 *N* gene mean Ct values from swabs and pools were not statistically different from one another ($p > 0.05$) for viral loads ranging from 10 to 66,779 GCE/PCR reaction (Fig 1B). Mean ΔCt values were < 0.5 cycle indicating that there is nearly complete quantitative absorption and recovery of clinical specimens from swabs using the vortex elution method.

Two additional dry swab elution methodologies, swab swirling within the tube and passive recovery, were compared to the vortex method to determine if mechanical disruption is necessary for full specimen recovery. Paired polyester swabs collected from healthy volunteers were spiked with a low-positive pool (2x LoD) and dry swabs were eluted using the vortex and swab swirling methods or the vortex and passive methods. Mean Ct values for the SARS-CoV-2 targets (*N*, *ORF1ab*, *S*) and human *RNase P* were all statistically higher ($p < 0.05$) for swirled and

**Table 1. Average swab dimensions and absorption volumes ± standard deviation (n = 10).**

| Swab | Average Length (mm) | Average Width (mm) | Average Absorption (μL) | |
|---|---|---|---|---|
| | | | Passive 10 sec | 5 Swirls in each direction |
| Puritan Foam | 15.4 ± 0.4 | 4.5 ± 0.2 | 28 ± 17.0 | 85 ± 7.9 |
| Fisherbrand Polyester | 13.7 ± 0.6 | 5.0 ± 0.1 | 121 ± 15.1 | 112 ± 8.8 |
| Copan Polyester | 15.3 ± 1.2 | 4.8 ± 0.1 | 113 ± 10.5 | 120 ± 13.8 |
| SteriPack Polyester (#60564) | 16.1 ± 0.9 | 5.2 ± 0.2 | 182 ± 16.3 | 184 ± 16.8 |
| SteriPack Polyester (#60567) | 15.1 ± 1.8 | 4.5 ± 0.1 | 120 ± 16.8 | 150 ± 21.4 |
| Copan FLOQ | 25.5 ± 0.2 | 3.8 ± 0.1 | 24 ± 3.8 | 64 ± 3.7 |
| Copan Mini-FLOQ | 15.5 ± 0.2 | 3.8 ± 0.2 | 50 ± 10.5 | 23 ± 2.2 |

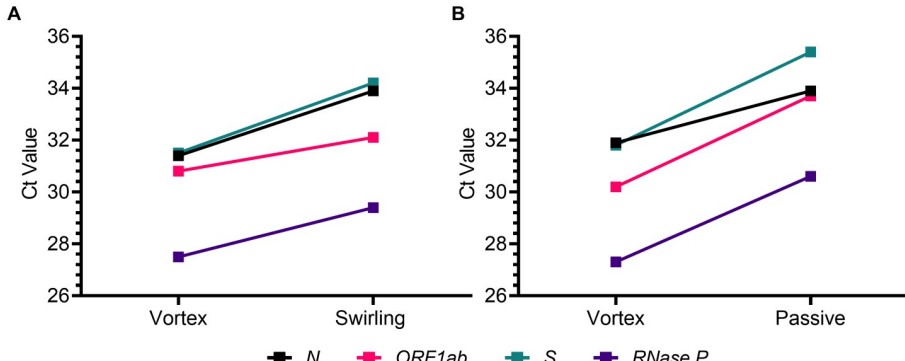

**Fig 2. Dry swab elution by vortex is superior to swab swirling and passive release.** Elution by vortex was compared to swab swirling (A) and passive release (B). Mean Ct values for the SARS-CoV-2 targets and *RNase P* are shown from paired polyester swabs collected from human volunteers (n = 5) and spiked with the low-positive hot pool (2x LoD; 20 GCE/PCR reaction). Ct values for swab swirling and passive release were all statistically higher (p<0.05) than Ct values from paired vortexed swabs.

passively eluted swabs than the paired vortexed swabs (Fig 2). Mean paired ΔCt values ranged from one to three cycles, correlating to a two- to seven-fold reduction in sample detection. These results indicate that mechanical disruption or agitation may be required for full elution of sample material from dry swabs.

## Polyester and foam swab surrogate specimen stability

Stability of surrogate specimens for RT-qPCR detection of SARS-CoV-2 was assessed with and without cold chain storage. Replicate polyester and foam swabs stored dry ("dry polyester" and "dry foam") and replicate polyester swabs stored in saline ("polyester saline") were prepared using low- or high-positive SARS-CoV-2 hot pools containing 22 and 144 GCE/PCR reaction, respectively. Ct values from the three swab types held under refrigerated conditions were not statistically different (p>0.05) through 72 hours (Fig 3A). Refrigerated dry polyester, dry foam, and polyester saline low- and high-positive replicates had 100% positive agreement, demonstrating stability through 72 hours. At elevated temperatures (40°C 12 hours, 32°C hold), all three swab types had 100% positive percent agreement for high-positive replicates through 72 hours. Ct values for low-positive dry polyester and dry foam replicates were not statistically different (p>0.05) through 72 hours, indicating that the two swab types were equivalent (Fig 3B). Viral RNA levels were relatively stable for dry polyester and dry foam swabs through 72 hours at both temperatures, with mean ΔCt values for *N* and *ORF1ab* < 3 cycles (less than 10-fold change in viral RNA levels). For both types of dry swabs, one of 20 low-positive replicates were inconclusive (only *N* detected) after 72 hours at elevated temperatures, resulting in 95% positive percent agreement, meeting the study acceptance criteria, and establishing stability for dry swab specimens through 72 hours. Stability for polyester saline surrogate specimens was established through 48 hours at elevated temperatures (100% positive agreement), however, Ct values were significantly higher (p<0.05) than dry polyester and foam after 48 hours. Seven of 20 low-positive polyester saline replicates were inconclusive (only *N* detected) after 72 hours at elevated temperatures for 65% positive percent agreement; thus, stability for polyester saline surrogate specimens was established through 48 hours at elevated temperatures (100% positive agreement).

Taken together, all three swab types (dry polyester, dry foam, and polyester in saline) were stable for up to 48 hours in the absence of cold chain. The data from these stability studies

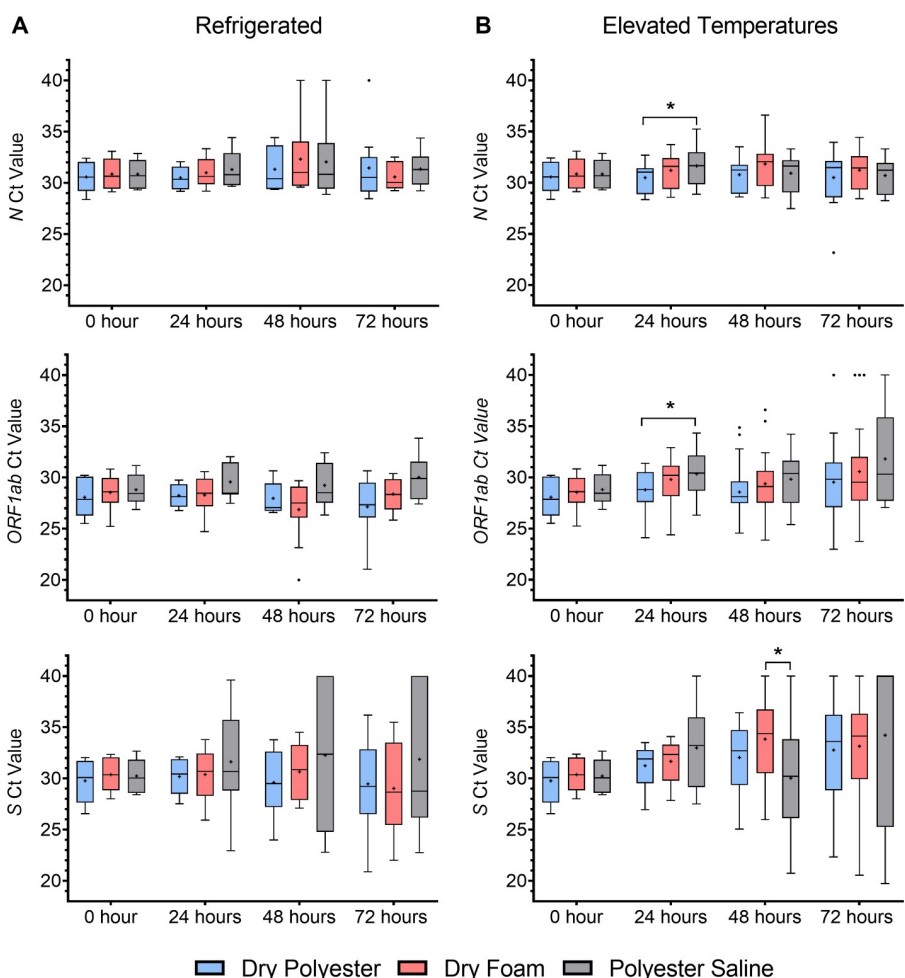

**Fig 3. Surrogate nasal swabs are stable at elevated temperatures mimicking warm-climate transport conditions.**
Dry polyester (blue), dry foam (red), and polyester swabs stored in saline (gray) were stored refrigerated (A, 4˚C) or at
elevated temperatures (B, 40˚C 12 hours, 32˚C) through 72 hours. No PCR amplification was assigned a Ct value of 40.
*Statistically different, p<0.05.

were provided to the FDA, resulting in the acceptance of the dry nasal swab (in addition to
polyester in saline) as a reliable specimen type for the detection of SARS-CoV-2 by RT-PCR.
The data were subsequently deposited in a file with the FDA, which allowed in vitro diagnos-
tics (IVD) manufacturers and laboratories to utilize the 48-hour stability claim for nasal swabs
without the requirement to replicate the studies. These data and conclusions have been refer-
enced by many IVD manufacturers and laboratories that have included home collections kits
for use with their tests.

## Polyester swab human matrix and clinical specimen stability

An additional evaluation of dry polyester surrogate specimen stability was performed using
matrix from individual human volunteers. Paired Copan and SteriPack #60564 polyester nasal
swabs were self-collected by healthy adults and inoculated with virus to create high-positive
(10x LoD) specimens. Consistent with the results obtained with surrogate specimens prepared
with porcine matrix, in the presence of human matrix, contrived dry polyester swabs were

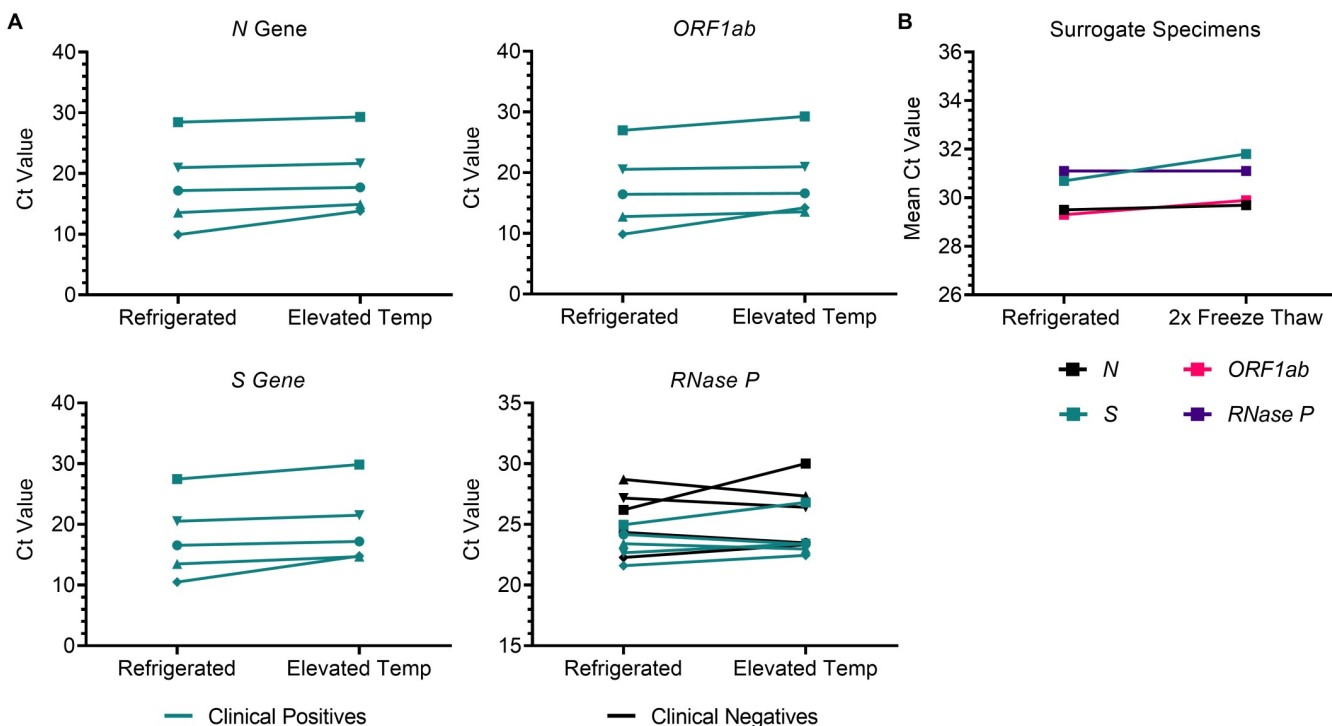

**Fig 4. Clinical and surrogate dry SARS-CoV-2 specimens are stable at temperatures that mimic warm- and cold-climate shipment.** (A) Paired collections from SARS-CoV-2 positive (n = 5; green) and negative (n = 5; black) patients were stored refrigerated and at elevated temperatures though 72 hours prior to RT-qPCR analysis for SARS-CoV-2 targets and *RNase P*. (B) Paired polyester swabs collected from human volunteers (n = 10) were spiked with the high-positive pool (10x LoD; 100 GCE/PCR reaction) and stored refrigerated or cycled through two freeze thaws.

stable for SARS-CoV-2 detection through 72 hours refrigerated and through at least 48 hours at elevated temperatures (the experiment was only conducted to 48 hours + 8 hours) (S1 Table).

The high temperature stability of dry surrogate specimens was confirmed with clinical specimens collected from five SARS-CoV-2 positive individuals and five healthy individuals using paired polyester swabs. Paired swabs were stored for 72 hours with one swab refrigerated and the other at elevated temperatures. Ct values for the three SARS-CoV-2 targets and *RNase P* were not statistically different (p>0.05) by storage temperature (Fig 4A), indicating that dry clinical specimens are stable through 72 hours at both refrigerated and elevated temperatures. Furthermore, dry surrogate specimens were stable for detection of SARS-CoV-2 and *RNAse P* for up to two freeze-thaw cycles (Fig 4B).

## Feasibility of home-based nasal specimen self-collection

The demonstrated stability of dry specimens at both low and high temperatures makes the dry polyester swab a suitable component of an at-home collection kit since it may be shipped stably year-round. The polyester (SteriPack #60567) dry swab is utilized in the HealthPulse@home COVID-19 Specimen Collection Kit for at-home, unsupervised self-collection of nasal specimens. Usability studies for the HealthPulse@home kit were performed in healthy adult (n = 34) and pediatric (n = 38) populations (demographics shown in S2 Table) to determine if individuals/caretakers could successfully self-collect nasal specimens at home without professional assistance and ship them in a dry tube within 80 hours for testing. Participants were recorded as they followed digital and/or paper instructions to complete the collection procedure. A study researcher assessed participant compliance to follow each step of the instructions

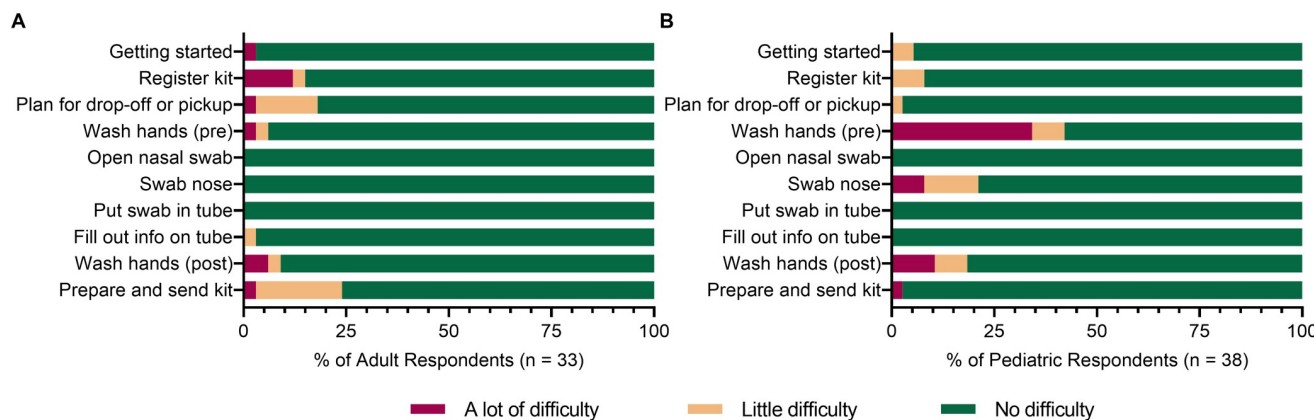

**Fig 5. Usability studies demonstrate feasibility of at-home nasal specimen collection.** Researcher's assessment of observed difficulty for each step during (A) adult and (B) pediatric usability studies for at-home, nasal specimen self-collection using the HealthPulse@home COVID-19 Specimen Collection Kit.

properly. Afterwards, participants were surveyed about their experience with the kit that included rating the amount of difficulty at each step of the procedure. In the adult study, which was performed first, subjects had no difficulty with the nasal swabbing procedure but did encounter some difficulty with kit registration, drop-off/pickup planning, and preparation for shipping (Fig 5A). Based on these results, the instructions were modified prior to the pediatric study, which alleviated the difficulty at these steps for pediatric caretakers (Fig 5B). Washing hands pre-collection caused "a lot of difficulty" for 34% of respondents in the pediatric study due to the parent not washing their hands before swabbing the child, or the child touching the supplies while the parent was away washing their hands. Swabbing only one of the child's nostrils resulted in "a lot of difficulty" for 8% of caretakers while not performing the four full rotations, inserting the swab deeper than needed, or the child pulling away resulted in "a little difficulty" for 13% of caretakers. The mean survey response for "Did you feel that you needed help while collecting your sample?" was 1.3 in both studies (scale: 1.0 = Disagree to 5.0 = Agree), indicating that the participants were confident in their self-collected specimens.

The majority of participants in both studies (82% adult; 71% pediatric) chose to schedule an online pickup, which did not require them to leave their home. The duration between specimen collection and delivery of each kit to the laboratory was tracked, except for one kit that was lost by an adult participant between drop-off at an in-office pickup site and pickup by the courier service. In the combined studies, 58% of participants' kits arrived at the laboratory within 24 hours, 30% arrived within 25–48 hours, 8% arrived within 49–80 hours, and 4% arrived after 80 hours. Upon inspection of returned kits in the laboratory, all adult (n = 33) and pediatric (n = 38) specimens were deemed suitable for testing. All self-collected specimens had detectable *RNase P* by RT-qPCR with Ct values ranging from 22.5–35.5 in the adult study and from 20.1–30.5 in the pediatric study. The predefined threshold (mean $\leq$ 32 cycles as required by the FDA) for nucleic acid sufficiency was met with mean *RNase P* Ct values of 23.4 and 25.1 in the adult and pediatric studies, respectively.

## Discussion

In the present study, we demonstrated that polyester swabs from multiple manufacturers are a suitable alternative to foam swabs for molecular detection of SARS-CoV-2. We evaluated the performance of polyester and foam nasal swabs stored in the absence of any transport media within a dry tube and developed a dry swab elution method utilizing PBS and a brief vortex,

which can be easily performed in the laboratory. Mechanical disruption by vortex resulted in nearly quantitative recovery of virus from dry swabs even at very low virus levels, while passive elution and swab swirling showed a marked reduction in viral recovery (although this may be sufficient in certain settings). We show that foam and polyester nasal swab specimens may be stored dry without transport media for up to 72 hours in the absence of cold chain. Polyester swabs stored in saline in the absence of cold chain were stable through 48 hours, however there was reduced stability of the low-positive (2x LoD) SARS-CoV-2 samples at 72 hours. Note that based on unpublished studies, buffered saline, such as PBS, may support extended stability beyond 48 hours. Our studies provided the FDA to allow the use of foam or polyester nasal swabs transported in saline, PBS, or dry tubes as a reliable specimen type for SARS-CoV-2 detection by RT-qPCR and this data has been made available via a Right of Reference.

Previous work determined that self-collected nasal specimens from the AN perform equivalently to nasopharyngeal and oropharyngeal specimens for SARS-CoV-2 detection by RT-qPCR [1, 2]. In this study, we demonstrate that very low viral loads near the assay limit of detection are detectable from nasal swab surrogate clinical specimens that are consistent with viral loads through 10 days from the start of COVID-19 symptoms (Y-P Tu, unpublished data). Viral infectivity studies conducted by the CDC (www.cdc.gov/coronavirus/2019-ncov/hcp/disposition-hospitalized-patients.html) and other groups have shown that specimens collected from confirmed SARS-CoV-2 patients with mild to moderate illness who are not severely immunocompromised and who are greater than 9–11 days from the start of symptoms do not yield culturable virus, thereby supporting the notion that nasal swab specimens are sensitive enough for detection of the virus through the infectious phase [6–8]. In the present study, we demonstrate the performance of AN swabs for SARS-CoV-2 testing using surrogate and clinical specimens and show that self-collected nasal swab specimens produce sufficient material for molecular testing as demonstrated by the reproducible detection of the *RNase P* human target. Importantly, the lower nasal swab self-collection instructions developed by Audere must be accurately followed to ensure a quality specimen for molecular testing (www.healthpulsenow.org/collect). These animated instructions are recommended by the FDA (https://www.fda.gov/medical-devices/letters-health-care-providers/recommendations-providing-clear-instructions-patients-who-self-collect-anterior-nares-nasal-sample) and have been made available via a Right of Reference. The usability studies for the HealthPulse@home COVID-19 Specimen Collection Kit demonstrate the real world feasibility of at-home, self-collection from children and adults using swabs transported dry in the absence of temperature controls. Participants indicated that they did not need help while collecting the specimen and all usability specimens had detectable *RNase P*, including the specimens that arrived at the laboratory after 80 hours and the specimens that were collected from only one nostril. The confusion regarding who should wash their hands in the pediatric study was addressed by modifying the collection instructions. The advantages of self-collected swabs from the AN, whether collected at home or in a healthcare setting, include improved patient tolerance, reduction of healthcare worker risk of infection, and preservation of PPE. It is anticipated that compliance with COVID-19 testing recommendations will increase with the use of a well-tolerated and simple collection device and will enable serial testing when appropriate.

This work demonstrates that dry nasal swabs are a reliable specimen type for SARS-CoV-2 detection, and usability studies performed in healthy individuals provides real-world feasibility of at-home, self-collected specimens. However, this study has several limitations. One limitation is the use of contrived swabs to assess specimen stability. While residual clinical specimens were used to mimic real-world specimens, an ideal study would have utilized patient specimens to assess SARS-CoV-2 specimen stability using dry polyester swabs and a reference method such as nasal or nasopharyngeal swabs stored in VTM. A second limitation of this

study is that the usability studies comprised of only healthy individuals and *RNase P* testing. Of note, one sample in the adult usability study had a *RNase P* value of 35.5 and while the overall study met the FDA defined mean $\leq 32$ cycle threshold, it is possible that this sample would fail SARS-CoV-2 detection. The high *RNase P* value is likely due to user variability which is a significant challenge in home collection. Further studies are needed to assess home-based nasal specimen self-collection, particularly in SARS-CoV-2 positive individuals, and to improve overall home collection processes.

The use of pre-positioned nasal swab collection kits in the home or workplace could facilitate more rapid COVID-19 testing and reduction in transmission. Differential diagnosis of other upper respiratory infections including influenza, which is also detectable in nasal specimens [9], would be feasible from a self-collected specimen. Nasal swab specimens are stable when transporting in dry tubes within 72 hours to the laboratory where they can be eluted using a robust and reproducible elution method. Studies to evaluate reduced elution volumes for increased assay sensitivity or direct swab elution into extraction or PCR buffer are under way. The utility of dry swabs can also be extended to future applications, including direct antigen testing for SARS-CoV-2. It is hoped that the inexpensive polyester nasal swab will bolster testing capabilities to improve patient outcomes specifically in low resource settings where swabs and other testing components are in limited supply.

## Supporting information

**S1 Text. Supporting materials and methods.**
(DOCX)

**S1 Fig. Polyester (poly) and foam nasal swab specimen stability study design.** Swabs were stored with refrigeration (A, control arm) and with extended periods of time at high temperatures (B, experimental arm). Swabs spiked with the pool of negatives (n = 2 per swab type) are not shown.
(TIF)

**S1 Table. Comparison of Copan and SteriPack (#60564) dry polyester swabs using human matrix to demonstrate comparability of performance and to validate stability.** Mean Ct and ΔCt values (stability time point– 0 h) ± standard deviation of the high-positive (10x LoD) pool in human matrix.
(DOCX)

**S2 Table. Respondent characteristics for the HealthPulse@home COVID-19 specimen collection kit usability studies (pediatric parent/guardian, n = 38; adult, n = 33).**
(DOCX)

## Acknowledgments

Many thanks to Karen Heichman (Bill & Melinda Gates Foundation) for many useful discussions. We thank Bob Setterquist (ThermoFisher), Thomas Bakken (University of Minnesota), Laura Bradner and Karen Harmon (Iowa State University), and Paul Isabelli (Audere) for providing materials and reagents. We thank the Everett Clinic-Part of Optum and UnitedHealth Group for clinical specimens.

## Author Contributions

**Conceptualization:** Leah R. Padgett, Yuan-Po Tu, Shawna D. Cooper, Douglas Rains.

**Data curation:** Leah R. Padgett, Douglas Rains.

**Formal analysis:** Leah R. Padgett.

**Investigation:** Leah R. Padgett, Lauren A. Kennington, Charlotte L. Ahls, Delini K. Samarasinghe.

**Resources:** Yuan-Po Tu, Shawna D. Cooper, James S. Elliott.

**Supervision:** Leah R. Padgett, Shawna D. Cooper, Douglas Rains.

**Writing – original draft:** Michelle L. Wallander.

**Writing – review & editing:** Leah R. Padgett, Michelle L. Wallander, Douglas Rains.

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
