## [Decision Letter · Decision Letter 0]

4 Feb 2021

PONE-D-21-00298

Polyester Nasal Swabs Collected in a Dry Tube are a Robust and Inexpensive, Minimal Self-Collection Kit for SARS-CoV-2 Testing

PLOS ONE

Dear Dr. Rains,

Thank you for submitting your manuscript to PLOS ONE. After careful consideration, we feel that it has merit but does not fully meet PLOS ONE’s publication criteria as it currently stands. Therefore, we invite you to submit a revised version of the manuscript that addresses the points raised during the review process.

The comments raised by two independent reviewers seem minor.

We look forward to receiving your revised manuscript.

Kind regards,

Etsuro Ito

Academic Editor

PLOS ONE

Journal Requirements:

2.We note that the grant information you provided in the ‘Funding Information’ and ‘Financial Disclosure’ sections do not match.

4.Thank you for stating the following in the Financial Disclosure section:

"Funding for this project was provided primarily by the Bill and Melinda Gates Foundation, through a direct award to Quantigen (D Rains, L. Padgett, L. Kennington, C. Ahls, D. Samarasinghe, and J. Elliott). Audere (S. Cooper), which conducted the usability study, used funds from the BMGF for their work, via a separate grant from the one provided to Quantigen. Finally, some reagents and consumables were provided free of charge to Quantigen by ThermoFisher.

gatesfoundation.org

thermofisher.com

A representative from BMGF did assist at various points in the design of the study. However, that individual did not participate in carrying out or analyzing the laboratory studies, nor did she contribute to the writing of the manuscript."

We note that one or more of the authors are employed by a commercial company: Quantigen Biosciences, The Everett Clinic-Part of Optum and Sciest, Mooresville, NC, United States

Reviewers' comments:

Reviewer's Responses to Questions

**Comments to the Author**

1. Is the manuscript technically sound, and do the data support the conclusions?

Reviewer #1: Yes

Reviewer #2: Yes

2. Has the statistical analysis been performed appropriately and rigorously? 

Reviewer #1: N/A

Reviewer #2: No

3. Have the authors made all data underlying the findings in their manuscript fully available?

Reviewer #1: Yes

Reviewer #2: Yes

4. Is the manuscript presented in an intelligible fashion and written in standard English?

Reviewer #1: Yes

Reviewer #2: Yes

5. Review Comments to the Author

Reviewer #1: Collection of specimens for SARS-CoV-2 testing was a significant challenge during the early phases of the covid19 pandemic, and seems likely to remain a challenge in many countries – particularly rural Africa and South America, for at least years. Although nasopharyngeal swabbing remains a “gold standard” sample for this diagnosis, mid-turbinate, nasal and saliva samples have largely replaced NP swabbing for initial diagnosis. Early studies on nasal samples were conducted using foam swabs (Tu et al., 2020), but limitations in production have required clinicians and laboratories to utilize whatever sampling materials were available.

In addition to challenges associated with the swab itself, transport has also been an issue. Laboratories have relied on VTM, UTM, saline (buffered and unbuffered), and dry swabs, without a strong evidence base for decision making. The use of dry swabs, if justified by the evidence, enables easier specimen transport, which is a “nice to have” in the countries with strong transportation infrastructure, but which is an absolute requirement in less developed countries.

Rains and coworkers have provided a useful and significant addition to the knowledge base by comparing sampling, transport and specimen elution for several swab types. The experimental approach, relying on the use of surrogate specimens, is a reasonable approach for initial investigation; the authors have used an appropriate approach and have described the work in a manner that can be replicated. Importantly, they have used viral load levels that are relatively near the limit of detection for the RT-PCR assay employed in their study. While one might have wished the authors to compare transport with PBS, VTM, UTM and MTM, as well as sterile saline, the decision to compare only with saline is reasonable, given the number of laboratories that have utilized saline as a transport medium, and the lack of data suggesting significant superiority of any other transport medium.

The inclusion of freeze-thaw stability studies and high temperature stability studies is particularly noteworthy, since both may apply to rural areas with extreme environments (Northern US and Canada, mountains, rural South America, US and Africa).

I have several rather minor critiques of the manuscript. It is not obvious to me that the average reader will understand the clinical importance of the Table 1 data on absorption – I recommend that the authors provide some explanation.

I also believe that this manuscript would benefit by a statement of limitations in the study. An ideal study would compare dry collection and transport for spun polyester with a reference technique, such as the nasal swab or NP swab method reported by Tu et al (Tu et al., 2020), using a similar study design. Such a study would provide more confidence in the results, and I think it would be worth pointing this out in the manuscript. I have not attempted to think through a comprehensive list of limitations, but I am sure that the authors would have others to add.

This is a significant contribution.

Tu, Y.-P. P., Jennings, R., Hart, B., Cangelosi, G. G. A., Wood, R. C., Wehber, K., Verma, P., Vojta, D., & Berke, E. M. (2020). Swabs collected by patients or health care workers for SARS-CoV-2 testing. The New England Journal of Medicine, 383(5), 494–496. https://doi.org/10.1056/NEJMc2016321

Reviewer #2: The manuscript describes a basic but well considered and informative study in establishing that dry nasal swab collection using polyester swabs are an acceptable method in order to support the need for alternative sampling methods given the paucity of FLOQ swabs and PPE, especially in the early phases of the COVID-19 pandemic. While the team did not use direct clinical samples, their methodology was robust and the key variants re swab types, elution methods, temperature variance and measurement of the RNAs was well drafted. I appreciated that they did not overextend their analyses on the samples but kept it relatively simple to understand.

I am bemused as to why the authors do not state that their work provided the data for the FDA to allow the use of dry nasal swabs as a reliable specimen type for SARS -CoV 2 Rt PCR, it is in the supplemental files. It is an important piece of work and they are too modest.

Comments I have that may improve the paper would be informing the reader as to why 32.5 cycles was the Ct threshold for RNase P detection and ergo sample acceptability? Allied to this is that later when assessing the user self-collection there is at least one sample that has a CT of 35.5 and yet the authors ‘pass’ this by taking the mean, which drops the RNase P below their threshold. It is not clear to me why the mean should be used as it reads as convenience to me. A sample collected with an RNaseP CT of 35.5 would fail for the SARS CoV-2 targets and this should be recognized, not masked by an application of the mean Ct. User error is a really big factor in why home testing is a real challenge even with something as simple as swabbing the nares. The authors detail logistical challenges re shipping within 80 hrs, per the 72 hr threshold they established, reason for the high RNase P sample should be discussed.

I found the abstract, introduction, M&Ms and results clearly written and relatively easy to follow. The discussion was written out of sequence in terms of which data and outcomes was discussed first. I would have preferred to read the discussion in the same order as the results, and not have the self-test component raised first.

Minor things. All materials used including software should be reference and in the same basic format. In this manuscript some products have catalogue #s, others don’t and GraphPad is offered by who?

6. PLOS authors have the option to publish the peer review history of their article (what does this mean?). If published, this will include your full peer review and any attached files.

Reviewer #1: **Yes: **Timothy Joseph O'Leary

Reviewer #2: No

---

## [Author Response · Author response to Decision Letter 0]

1 Mar 2021

We have included a comprehensive response to reviewers in a separate document entitled "response to reviewers." In addition, we have made the requested changes to both the manuscript and to supporting figures, including all style- and content-related issues from the original submission. In our emended cover letter, we have provided the appropriate statements regarding financial disclosures and commercial affiliations.

---

## [Decision Letter · Decision Letter 1]

10 Mar 2021

Polyester Nasal Swabs Collected in a Dry Tube are a Robust and Inexpensive, Minimal Self-Collection Kit for SARS-CoV-2 Testing

PONE-D-21-00298R1

Dear Dr. Rains,

We’re pleased to inform you that your manuscript has been judged scientifically suitable for publication and will be formally accepted for publication once it meets all outstanding technical requirements.

Kind regards,

Etsuro Ito

Academic Editor

PLOS ONE

Reviewers' comments:

Reviewer's Responses to Questions

**Comments to the Author**

1. If the authors have adequately addressed your comments raised in a previous round of review and you feel that this manuscript is now acceptable for publication, you may indicate that here to bypass the “Comments to the Author” section, enter your conflict of interest statement in the “Confidential to Editor” section, and submit your "Accept" recommendation.

Reviewer #1: All comments have been addressed

2. Is the manuscript technically sound, and do the data support the conclusions?

Reviewer #1: Yes

3. Has the statistical analysis been performed appropriately and rigorously? 

Reviewer #1: Yes

4. Have the authors made all data underlying the findings in their manuscript fully available?

Reviewer #1: Yes

5. Is the manuscript presented in an intelligible fashion and written in standard English?

Reviewer #1: Yes

6. Review Comments to the Author

Reviewer #1: My comments have been adequately addressed, and I beleive that this paper is ready for publication. I appreciate the addiitonal comments regarding swab absorption, and the atatement of limitations. Thank you.

7. PLOS authors have the option to publish the peer review history of their article (what does this mean?). If published, this will include your full peer review and any attached files.

Reviewer #1: No

---

## [Editor Report · Acceptance letter]

5 Apr 2021

PONE-D-21-00298R1 

Polyester nasal swabs collected in a dry tube are a robust and inexpensive, minimal self-collection kit for SARS-CoV-2 testing 

Dear Dr. Rains:

I'm pleased to inform you that your manuscript has been deemed suitable for publication in PLOS ONE. Congratulations! Your manuscript is now with our production department. 

Kind regards, 

on behalf of

Prof. Etsuro Ito 

Academic Editor

PLOS ONE